# Improving co-registration of geoscientific imaging datasets with micro-sized marker structures on rock samples.

Rosa A. de Boer<sup>1</sup>, Daan H. Wielens<sup>2</sup>, and Lennart V. de Groot<sup>1</sup>

Correspondence: Rosa A. de Boer (r.a.deboer1@uu.nl)

**Abstract.** Polished geological samples are frequently used in geoscientific research to investigate the chemical and physical characteristics of rocks. A broad range of imaging techniques is available to analyze such samples, but when combining datasets from multiple imaging techniques, an accurate co-registration of the datasets is often challenging. In this study, we investigate this issue in the context of Micromagnetic Tomography (MMT; De Groot et al., 2018, 2021). MMT combines surface magnetometry data with computed tomography (CT) data to analyze the magnetic state of rock samples. By combining the spatial (position) and dimensional (size) information of the magnetic grains in the samples with their magnetic surface expression, the individual magnetic moments per grain can be determined. This information can be used for paleomagnetic and rock-magnetic studies. Calculating the magnetic moments of the grains strongly depends on the correct co-registration of the two datasets, which proves to be challenging. In this study, we used two test samples for the application of micro-sized marker structures, to further develop the methodology of MMT. The marker structures are applied by microlithography and Nb-sputter coating, which are standard techniques used in the semiconductor industry. We determined that the marker structure application is possible on typical MMT samples. Marker structures larger than ca. 10  $\mu$ m are clearly visible under the Quantum Diamond Microscope (QDM) used for the surface magnetometry. Given a sufficient marker structure thickness, they can also be observed in the CT scans used for determining the positions and shapes of the magnetic carriers. The marker structures are useful for identifying the orientation and location of the samples during measurements and can be used for scaling and mapping of the two datasets during data processing. Nb-marker structures do not fluoresce under the ODM, which means that no magnetic interference occurs during measurements. The application procedure is time-consuming but is valuable when a sample is lacking natural marker features, it makes the data processing time in MMT significantly faster, and more precise. This method can be useful for MMT, for Quantum Diamond Microscopy in general, and for broader geological applications that require visible anchor points for sample placement or marker structures for the co-registration of multiple datasets.

#### 1 Introduction

In geoscience research it is common to combine datasets from multiple measurement methods to gain an understanding of complex Earth systems. For example, different imaging techniques may be used to characterize the physical and/or chemical properties of Earth materials. However, combining and aligning datasets from multiple imaging techniques can be technically

<sup>&</sup>lt;sup>1</sup>Paleomagnetic laboratory Fort Hoofddijk, Department of Earth Sciences, Utrecht University, The Netherlands.

<sup>&</sup>lt;sup>2</sup>MESA<sup>+</sup> Institute for Nanotechnology, University of Twente, Enschede, 7522 NB, The Netherlands.

**Figure 1.** Example of MMT data, after Cortés-Ortuño et al. (2021). The blue and red surface scan shows the magnetic surface signal of the sample. The shape, size, and location of the magnetic sources in the sample are obtained with three-dimensional imaging.

challenging, particularly when a high spatial precision is pivotal to the outcome. In this study, we focus on the case of Micromagnetic Tomography (MMT), a novel technique in rock-magnetism and paleomagnetism that is used to reconstruct the history of Earth's magnetic field from polished rock samples (De Groot et al., 2018, 2021). MMT combines magnetic scans with three-dimensional imaging of magnetic sources in a sample (Fig. 1), which presents practical challenges. One of the major hurdles in the MMT workflow is the co-registration of these two types of imaging data, needed for a meaningful and accurate result. This paper explores a potential solution for aligning the datasets with high precision through the application of micro-sized marker structures on the rock samples used for MMT.

MMT combines two complementary datasets, integrating surface magnetometry with spatial (position) and dimensional (size) data of magnetic grains in rock samples. With this combined information, the magnetic moments of the individual magnetic grains can be calculated (De Groot et al., 2018, 2021). A Quantum Diamond Microscope (QDM) is used to obtain the surface magnetic flux density and a micro- or nanoCT scanner is used to obtain the spatial and dimensional data (Fig. 2). The co-registration of these data sets is cumbersome (Kosters et al., 2023), due to discrepancies in the resolution between the QDM and CT scanner. Although both instruments have known resolutions, their spatial scale is often not directly compatible: one micrometer in a QDM image does not necessarily correspond to one micrometer in a CT image. This is likely due to the calibration differences between the two machines and the data formats they produce. Nevertheless, an accurate mapping between the magnetic surface scans and micro- or nanoCT data is paramount for reliable MMT results because the accuracy of the calculated magnetic moments strongly depends on this.

Additionally, we encountered other problems that complicate the co-registration process or sample handling in general. Typical MMT samples are thin or thick slices of volcanic or sometimes sedimentary rock containing magnetic grains. The grain size of the minerals in these rock samples can be so small that we cannot easily recognize the grains on micrographs created with the QDM. Since matching these data with the CT scans is done optically, this complicates the co-registration process. Another issue can arise when a sample is very dark, for example, when a thick slice of rock is used for measurements. It is

difficult to identify magnetic grains optically with the QDM if this is the case, again complicating the co-registration process. Another issue occurs when the magnetic grain density in the samples is large, and the grain boundaries are not distinguishable on the micrographs created with the QDM. If the sample has a high magnetic grain density, and particularly if the grains are very homogeneous in size and shape, it is difficult to recognize them and to align the datasets optically. Problems can also arise depending on the type of data used in MMT. Sometimes, the pixel size is poorly calibrated or unknown in one of the scans, depending on where and how the measurements were done, which means that scaling the two datasets can be complicated. Lastly, if samples are measured remotely, it can be complicated for the operator to recognize the exact area of interest in a sample if the sample lacks distinct or recognizable features.

These problems led us to explore options for applying marker structures on rock samples to enable the tracking of the exact location and dimensions of our samples during measurements. Micro- and nano-sized patterns can be deposited on the surface of a rock sample with a particular design or structure. This approach ensures accurate scaling and co-registration of the data, and it can ease the sample placement during measurements. If anchor points are visible during both types of analysis, they can be used to correlate the data with the use of QDMlab software (Volk et al., 2022) or with 3D-visualization and analysis software. A standard technique for applying marker structures in the semiconductor industry is UV-microlithography (e.g. Mack, 2006; Razeghi, 2019). This technique is rarely applied to rock samples as it is challenging to apply to heterogeneous material, but an example can be found in Quintanilla-Terminel et al. (2017), and a similar use is described in Allais et al. (1994). Here, we test this method to determine whether it is suitable for use in Micromagnetic Tomography. We tested the ease of applying this process to our specific type of samples and whether the marker structures are easily distinguishable in both the magnetic surface scanning and micro- or nanoCT scans. Non-magnetic materials can sometimes fluoresce in the QDM, which disturbs the magnetic measurements. Therefore, we also tested whether the applied metal sputter coating interfered with our measurements. We applied the marker structures to volcanic samples from Réunion Island. Since volcanics are the most commonly used type of sample in MMT, we consider these an appropriate test subject to improve the accuracy and ease of co-registration in MMT.

# 70 2 Methodology

55

The techniques described below for applying marker structures on samples are based on standard techniques widely used in the semiconductor industry (e.g. Mack, 2006; Razeghi, 2019). The procedure requires equipment which is usually part of cleanroom laboratories used for the fabrication of solid-state devices or nanotechnological research. Given the limited use of these techniques in Earth Sciences, we provide a broad description of these techniques here. Fig. 4 shows a schematic overview of the steps involved in applying the marker structures. The procedure was carried out at the MESA<sup>+</sup> Institute for Nanotechnology at the University of Twente (The Netherlands).

**Figure 2.** Example images illustrating the data used in Micromagnetic Tomography. a) and b) are data obtained with the QDM. a) is a LED micrograph of the sample. b) is a vertical component (Bz-) map of the magnetic surface flux. c) is density-based data obtained with the nanoCT scanner. d) is magnetic grain data extracted from the CT data with the use of 3D-visualization and analysis software. One QDM pixel equals 1.20 μm and the field of view equals 2.30 mm x 1.44 mm. All images are co-registered and scaled to figure a).

# 2.1 Sample preparation

85

## 2.1.1 MMT sample preparation

Typical MMT samples are tiny drill cores from petrographic thin sections or thick sections (Fig. 3). The diameter of the samples depends on the research purpose; a smaller sample diameter allows for higher resolution nanoCT scanning. However, sample preparation becomes increasingly difficult as the sample diameter decreases, making 1.2 mm a practical minimum diameter. For the current study a diameter of 5 mm was chosen, to ease the process of sample preparation and the marker structure application. Two samples were created, one was drilled from a 50  $\mu$ m thick section (sample RE5) and the other sample was drilled from a 30  $\mu$ m thin section (sample RE25).

For the application of marker structures, it is important to minimize the surface roughness of the samples. Both the thin and thick section were polished to a standard polished surface finish before drilling the MMT samples from them. After the drilling procedure, the samples were polished using a colloid silica suspension, resulting in a surface roughness of approximately 50

#### Diameter 1.2 - 5 mm

Figure 3. Schematic of the typical geometry of an MMT sample. The samples are drilled from a petrographic thin section (30  $\mu$ m) or thick section (50  $\mu$ m) and their diameter depends on the research purpose and corresponding desired CT resolution. For the current study, we drilled samples with a diameter of approximately 5 mm from both a thin section and a thick section.

nm, depending on porosity, defects, and mineralogy in the rock samples. This polishing method is also used to remove any surface magnetization that may have been induced during the standard thin/thick section polishing (De Groot et al., 2014).

A uniform thickness of all samples is necessary for treating multiple samples during the marker structure application procedure, and the sample holder accommodates a maximum sample thickness of a few mm. Furthermore, it is important to ensure that the adhesive used for thin/thick section preparation is resistant to acetone, as this is used as a solvent during the marker structure application procedure. Another important consideration is that the samples are subjected to a 

**Figure 4.** Schematic of the application of marker structures on a rock sample, by means of UV-lithography and sputter coating. a) sideview of a typical MMT sample consisting of a tiny drill core from a thin section. b) application of photo resist on the sample. c) a mask with the desired structures is placed above the sample. UV-light shines through the transparent parts of the mask and breaks down the exposed photo resist. d) the exposed photo resist is easily removed after developing. e) A Nb-sputter coating is applied on top of the sample. f) lift-off; the sample is placed in acetone for 30 minutes to remove all the unwanted sputter coating. The desired structures are left in Nb on top of the sample.

Last, the samples are developed by immersing them in a chemical solution that dissolves short polymer chains much faster than long polymer chains, meaning that the unexposed resist remains on the sample surface while the exposed resist is removed (Fig. 4d). For this purpose, Fujifilm OPD 4262 is used for 1 minute. Once developed, the samples are immersed in water twice for 30 seconds each, to remove the developer and stop the process.

## 115 2.1.3 Sputter coating

After the UV-lithography, the samples are loaded in the vacuum chamber of a sputtering system. Before the deposition of Nb, the samples are etched by Ar-plasma to remove any contaminants from the sample surface. The etching is carried out at 250 W for 30 seconds.

Then, the samples are sputtered with a thin film of Nb, although another non-magnetic metal could be chosen as well for the purpose of MMT. A coating thickness of approximately 105 nm was achieved by sputtering at a radio-frequency power of 250 W for a duration of 70 seconds (Fig. 4e). Although this thickness is below the nanoCT detection limit, we still expect to be able to see the coating on the CT scans. The significant density contrast between the coating (ca. 8.5 g/cm<sup>3</sup>) and the minerals to which it is applied (ca. 2.5-5.5 g/cm<sup>3</sup>) is expected to result in combined densities that exceed those of the minerals in the samples. These anomalies will indicate where the coating has been deposited on the samples.

#### 125 **2.1.4** Lift-off

130

After the sputtering procedure, the entire sample surface is coated with a metallic layer. Part of this coating is deposited on top of the photo resist and part of it is directly deposited on the sample, where the photo resist was previously dissolved. The metal that is not desired needs to be removed, so only the desired structures are left on the sample surface (Fig. 4f). Therefore, we dissolve the resist and thereby remove its top layer of metal in acetone for 30 minutes, leaving only the Nb that was deposited onto the sample surface. Afterwards the samples are sprayed with acetone to remove the final leftovers of resist and metal. The sample surface is then cleaned with ethanol and dried with a nitrogen spray gun.

## 2.2 Micromagnetic Tomography

MMT combines surface magnetometry measurements of magnetic grains in a rock sample with the spatial and dimensional data of those magnetic grains (De Groot et al., 2018, 2021). These combined data can be used for inverse modeling to determine the magnetic moment per individual magnetic grain (Fabian and de Groot, 2019).

For this study, the two different data sets are used solely to demonstrate the use of marker structures in the co-registration process. The calculation of magnetic moments of individual magnetic grains is beyond the scope of this study.

# 2.2.1 Quantum Diamond Microscopy

The QDM at the Institute for Rock Magnetism of the University of Minnesota (United States of America) is used to map the magnetic flux density of the samples. Their laboratory set-up is similar to the set-up developed by Glenn et al. (2017). The

Table 1. QDM measurement settings.

| Settings                     | RE5    | RE25   |
|------------------------------|--------|--------|
| Exposure time ( $\mu$ s)     | 28,000 | 29,000 |
| Camera frame time ( $\mu$ s) | 28,059 | 29,059 |
| Microwave power (dBm)        | -25    | -25    |
| Number of sweeps             | 20     | 20     |
| Start frequency 1 (GHz)      | 2.837  | 2.837  |
| End frequency 1 (GHz)        | 2.849  | 2.849  |
| Start frequency 2 (GHz)      | 2.892  | 2.892  |
| End frequency 2 (GHz)        | 2.904  | 2.904  |
| Frame pairs per frequency    | 20     | 25     |

magnetic data is acquired from an optical image with a maximum field of view of 1,920 x 1,200 pixels and a spatial resolution of 1.20  $\mu$ m. A 0.9 mT bias field is applied during the measurement, with a continuously switching polarity. Technical details of this measurement method can be found in Farchi et al. (2017); Fu et al. (2020); Glenn et al. (2017), and Levine et al. (2019). The strengths and limitations of the wide-field QDM are described in-depth in Scholten et al. (2021). Reflected light microscopy is incorporated in the QDM system to link the magnetic field maps with the mineralogy of the samples. The QDM settings for the measurements of the current study are listed in Tab. 1. A global fluorescence correction of 0.2 is applied (Fu et al., 2020; Volk et al., 2022), the results are unbinned, and the magnetic map is converted to a map with the vertical component (Bz) of the magnetic field (Fu et al., 2020; Lima and Weiss, 2016).

# 2.2.2 Nano-Computed Tomography

The Multiscale X-ray NanoCT SkyScan2211 at the Faculty of Dentistry, University of Oslo is used to obtain the spatial and dimensional information of the magnetic grains in the samples. This is a non-destructive technique that can be used to characterize geological specimens (Cnudde and Boone, 2013; Martini et al., 2021). NRecon software is used for corrections of the raw data and to convert the CT scan projection images to cross-section images. NanoCT scans produce a three-dimensional image of the X-ray attenuation contrast in a sample. This can be interpreted as density variations in a sample (Cnudde and Boone, 2013; Jussiani and Appoloni, 2015; Lima and Jussiani, 2024; Sakellariou et al., 2004). Typical volcanic MMT samples consist of silicate minerals and, often magnetic, iron oxides. The large density difference between the silicate minerals (ca. 2.5-3.5 g/cm³) and the iron oxides (ca. 4.5-5.5 g/cm³) in the samples is clearly distinguishable in the nanoCT scans as a bimodal attenuation spectrum. It is therefore possible to precisely locate the iron oxides with sizes above the pixel size and to determine their shape and volume. The image processing is done with Dragonfly software. First, median filtering is applied for noise reduction and then K-means filtering is applied to categorize the data into groups based on their density contrast. The minimum value separating the high-density peaks of the iron oxides from the lower density matrix is used as a threshold. With

Table 2. NanoCT measurement and correction settings.

| Settings                      | RE5   | RE25  |
|-------------------------------|-------|-------|
| Source voltage (kV)           | 95    | 95    |
| Source current $(\mu A)$      | 200   | 200   |
| Exposure (ms)                 | 2,300 | 2,100 |
| Image pixel size $(\mu m)$    | 1.38  | 1.38  |
| Number of projections         | 1,895 | 1,895 |
| Ring artifact correction      | 9     | 9     |
| Beam hardening correction (%) | 50    | 50    |

this, the iron oxide locations and shapes can be extracted from the CT data. The threshold is set at the minimum value of the high-density group to ensure that all magnetic iron oxide grains are identified. The measurement and data correction settings used in the current study are listed in Tab. 2.

## 165 2.2.3 Co-registration

To carry out an inversion, the data sets from both the QDM and nanoCT have to be co-registered in the same coordinate system. This ensures that the magnetizations determined with surface magnetometry are attributed to the corresponding magnetic grains in the sample. Typically, this is done by optically scaling the nanoCT data to reflected LED light images of the sample surface made with the QDM, by matching the geometry of the surface grains in both data sets. Since the LED images of the QDM share their coordinate system with the magnetic surface maps made with the QDM, the nanoCT data are inherently co-registered to the latter as well. This work provides results on experiments with the application of marker structures on the sample surface, to ease the co-registration process.

#### 3 Results

We have successfully applied micro-sized marker structures to our two MMT samples, the applied structures are shown in Fig. 5. A micrograph shows exposed photo resist (Fig. 5a) with the largest structures applied to the samples, two arrows and a line with a 50-60 μm width. In Fig. 5b the smaller applied structures are shown in a micrograph, after completion of the procedure. The numbers are 20 μm tall and the line width is 3 μm. The dots make up a grid with distances of 100 μm and are 3 μm in diameter. The marker structures are clearly visible in the LED light of the QDM (Fig. 5c, d), down to a size of approximately 10 μm. The markers do not fluoresce under the QDM's laser light, meaning no magnetic interference occurs during measurements (Fig. 5e, f). Although the thickness of the Nb-layer (105 nm) of the marker structures is below the minimum pixel size of the CT scanner (1.38 μm), the density difference with the surrounding minerals is large enough to see them in the CT scans (Fig. 5g, h).

Figure 5. Images of the applied marker structures. One QDM pixel equals 1.20  $\mu$ m and the field of view equals 2.30 mm x 1.44 mm. All images except b) are co-registered and scaled to figures c) and d). a) is a micrograph of the applied photo resist, already exposed to UV-light. The large marker structures of two arrows and a line are visible in the photo resist on the sample surface. b) micrograph of the smallest applied marker structures on the sample surface, after lift-off. The white dots are 3  $\mu$ m in size, the white numbers are 20  $\mu$ m tall. c) and d) are LED micrographs of the applied markers on the sample surface as observed under the QDM. e) and f) are the Bz-maps of the magnetic surface scans measured with the QDM. The marker structures do not fluoresce under laser light, meaning that they do not interfere with the magnetic signal of the sample. g) and h) are the density-based data obtained with the nanoCT scanner. Only the larger marker structures are visible.

We tried the marker structure application procedure on six test samples. From these test samples, only one sample was unsuccessful because the bottom surface of the sample was too rough to form a proper vacuum seal in the spin coater. This bottom surface sometimes becomes damaged during the drilling of the tiny cores from the thin sections. This can easily be fixed in the future by polishing the bottom of the samples before usage. It is important to consider that the thickness of all samples should be similar, for sample mounting purposes. All other test samples were successfully used for marker structure application.

## 4 Discussion

## 4.1 Micro-marker structure application

The marker structure application process described in this study is not typically used on geological samples, but it can be applied to petrographic thin sections successfully (Fig. 5). Although the samples have a suboptimal surface roughness (>>10 nm), the photo resist adheres well to the sample surface. A little build-up of photo resist is present on the edges of the samples during the process, but this does not interfere with the further marker structure application. The Nb-sputter coating adheres sufficiently to the exposed sample surface as well. Other non-magnetic metal coatings can be used, if Nb is not available or otherwise impractical to use.

The marker structures look more irregular than usual for the semiconductor applications of this technique. However, the size of the marker structures corresponds to the size of the applied masks. Therefore, the structures can be used for scaling purposes in MMT or other imaging techniques. Little limitations exist regarding the type and the size of the pattern used on the samples.

The Nb-sputter coating was applied with a thickness of 105 nm. A thicker layer was not attempted in this test case, because thicker layers can cause problems during the lift-off phase of the procedure. For example, the structures can collapse or detach from the sample surface. Ideally, we would like to apply a Nb-layer of >400 nm to ensure their visibility in the nanoCT scans, because ca. 400 nm is the CT resolution achieved for standard MMT samples with a diameter of 1.2-1.5 mm. Although this is a significant increase from the current Nb-layer thickness, we encountered no issues during the lift-off process. Therefore, we expect that thicker marker structures can be successfully applied for future purposes with minor procedure optimization.

It is worth noting that the experimental procedure for applying the marker structures to six samples (maximum amount in current set-up) costs several hours and is both labor intensive and technologically demanding. While this technique is valuable when needed, it is not recommended unless it is essential. When many samples are required, it decreases the time of the procedure per sample significantly, as the samples can be treated in batches.

#### 210 4.2 Micro-marker structure visibility

The marker structures are visible under LED illumination, making them useful for sample handling and placement of the sample under the QDM sensor. Since the marker structures do not fluoresce and therefore cause no magnetic interference, they

are well-suited for MMT applications and for Quantum Diamond Microscopy in general. Additionally, the markers are visible in the CT scans, allowing for co-registering the CT images with the QDM images.

For this test study, the samples were cut to a diameter of 5 mm. As a result, the CT resolution was relatively low at 1.38  $\mu$ m. Typically we use samples with a 1.2-1.5 mm diameter for MMT studies, which means we can achieve CT resolutions of ca. 400 nm. In this test study we can only see the larger marker structures of 50  $\mu$ m wide and 105 nm thick (Fig. 5a) in the CT images but we cannot see the smaller marker structures of max. 20  $\mu$ m tall, 3  $\mu$ m wide, and 105 nm thick (Fig. 5b). When the samples are cut to their regular size, we expect that we can see marker structures of a few  $\mu$ m in diameter in the CT scans, due to a better scanning resolution.

The co-registration can be successfully achieved with the applied marker structures with a 50  $\mu$ m width. When multiple structures are applied at known distances from one another, they provide a highly reliable reference system for aligning the datasets. The main improvement is that the co-registration process no longer depends on the presence of natural features of the samples that are recognizable in both types of imaging. Instead, the marker structures provide a consistent reference, independent of the sample's properties. This ensures that co-registration is possible for every sample, including those where natural, recognizable features are insufficient. Such cases include samples with tiny grain sizes that are difficult to distinguish optically, samples that appear too dark in the LED image of the QDM, or samples with high magnetic mineral density, obscuring the identification of any distinct features.

## 4.3 Other methods

230 Microlithography is a useful and feasible method for MMT, Quantum Diamond Microscopy in general, and other methods involving petrographic thin sections or geological samples. However, there are other techniques available that might be more suitable, depending on the sample type or research purpose of a certain study. Some examples are provided below; however, this is not an exhaustive list. Since micro- and nanoscale measurement techniques are becoming more common within Earth Sciences, undoubtedly new techniques and improvements will emerge for application on geological samples in the near future.

A simple technique such as paint sprayed on samples, producing droplets with a diameter of 10-100  $\mu$ m (Dautriat et al., 2011) can be used to co-register images optically. This approach is only viable for MMT if the density difference between the paint and the minerals in the samples is sufficiently large for the droplets to be observable in the CT scans. Furthermore, depending on the thickness of the droplets, the stand-off distance between the sample and the sensor of the QDM increases. Because magnetic flux density decreases exponentially with distance from the sensor, this increased stand-off distance can reduce the signal strength below the QDM's detection limit. As a result, this technique can only be used in MMT when applied on strongly magnetic samples.

Another technique consists of high-temperature (450°C) annealing of a metallic thin film applied with standard surface metallization. This results in the formation of micron-sized metallic droplets on the samples surface (Bourcier et al., 2013). When the annealing is carried out at lower temperatures (220°C) the size of the metallic droplets decreases to nanoscale (Parlangeau et al., 2022). The advantage of this method compared to microlithography is its relatively low labor intensity. However, this technique can only be applied to MMT samples if thermal demagnetization does not interfere with the research

purpose. Heating a sample can alter or erase its natural remanent magnetization (NRM), which is often the signal of interest in paleomagnetic studies. Therefore, this method is only suitable when preserving the original magnetic state is not essential. It is important to note that the type of metal applied to the samples might fluoresce under the QDM's laser light even if it is non-magnetic. Therefore, this should be tested prior to application in MMT to ensure no magnetic interference occurs during measurements.

Another option would be to engrave fiducial markers in the edge of the samples with the use of a FIBSEM, similar to the process described in Out et al. (2024). This process damages the samples, so it can only be applied to an area outside the region of interest for an MMT study. If the fiducial markers are sufficiently large, they can be used for optical placement of the samples in the QDM and for the co-registration of the images of the QDM and the CT scans.

## 5 Conclusion

The precise co-registration of multiple imaging datasets remains a challenge in geoscience research. In this study, we demonstrate that microlithography provides a solution for co-registering imaging datasets. In the context of MMT, the application of micro-sized marker structures on typical MMT samples is possible and feasible for overcoming co-registration challenges between surface magnetometry and 3D-imaging. Despite the relatively large surface roughness of polished geological samples, the marker structures can be applied with high precision and with accurate dimensions. A marker structure size of >10  $\mu$ m is recommended to ensure visibility in both the QDM images as well as the CT scans. Although a thickness of 105 nm is sufficient for visualization in the CT scanner, preferably this thickness would be increased towards a few hundred nanometers to ensure visibility in CT scans. Since the dimensions of the marker structures are known with great precision, they can be used for scaling purposes as well as co-registration of datasets. The visibility of the marker structures under the LED light of the QDM may ease the sample placement process, in case of a specific area of interest that needs to be measured if a sample is lacking natural markers. Furthermore, the applied Nb-coating does not fluoresce under the laser light of the QDM and does therefore not interfere with the magnetic measurements. Although the marker structure application procedure is labor intensive and time consuming, it is useful for MMT purposes, Quantum Diamond Microscopy in general and in other applications involving polished geological samples.

Author contributions. RAdB is responsible for writing the original draft, visualization, data curation, formal analysis, and part of the investigation. DHW is responsible for reviewing and editing the manuscript, as well as part of the investigation. LVdG is responsible for the conceptualization, funding acquisition, and supervision (CRediT Taxonomy).

Competing interests. The authors declare that they have no conflict of interest.

Acknowledgements. We thank the Institute for Rock Magnetism of the University of Minnesota for providing RAdB with a Magnetic Microscopy Fellowship to carry out QDM measurements. This project has received funding from the European Union's Horizon 2020 research and innovation program (Grant agreement No. 101005611) for Transnational Access to RAdB conducted at the Department of Biomaterials, University of Oslo. We extend our thanks to Liebert Parreiras Nogueira and Torben Hildebrand for their expert assistance with the nanoCT measurements. This project has received funding from the European Research Council (ERC) under the European Union's Horizon 2020 research and innovation program (Grant agreement No. 851460 to LVdG). The authors used ChatGPT (OpenAI) for text editing in the preparation of this manuscript. All intellectual contributions, including the research design, data analysis, and conclusions, are solely those of the authors.

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
