# Peer review of "Improving co-registration of geoscientific imaging datasets with micro-sized marker structures on rock samples."

_EGUsphere, 2025_

## Author Response (AR1)

**Response to reviewers**

We would like to thank the reviewers for their comments and recommendations on our manuscript. The reviewers raise a number of valid points (in black) that we addressed (in blue) in the main text (in italics) as follows:

**RC1**

L6 and L32: What is the difference between spatial and dimensional information? Does space not include all three dimensions? This phrasing was used to discern between the location (spatial information) of the grains and the size (dimensional information) of the grains. This has been clarified in the text.

"By combining the spatial (position) and dimensional (volume) information of the magnetic grains in the samples with their magnetic surface expression, ..."

Fig. 1 and L27f: During the first reading it was not clear where co-registration would be necessary at this stage, since MMT and imaging of magnetic sources might be the same procedure. I would suggest combining this sentence from the abstract ("MMT combines surface magnetometry with computed tomography (CT) to analyze the magnetic state of rock samples.") with the main text of the introduction (lines 27f: "MMT combines magnetic scans with three-dimensional imaging of magnetic sources in a sample ..." This has been changed to 'surface magnetometry data with CT data' to clarify that two datasets are being used.

"MMT combines surface magnetometry data with computed tomography (CT) data to analyze the magnetic state of rock samples"

Fig. 2: It is a bit hard to understand the caption. Put the two sentences regarding the scale at the end of b) or at the end of the figure caption. What sample is it (RE5, RE25 or a different)? The sentence about scaling has been adjusted. The sample shown is an example image from a different study, used to illustrate the type of data used in MMT.

"Example images illustrating the data used in Micromagnetic Tomography. a) and b) are data obtained with the QDM. a) is a LED micrograph of the sample. b) is a vertical component (Bz-) map of the magnetic surface flux. c) is density-based data obtained with the nanoCT scanner. d) is magnetic grain data extracted from the CT data with the use of 3D-visualization and analysis software. One QDM pixel equals 1.20  $\mu$ m and the field of view equals 2.30 mm x 1.44 mm. All images are coregistered and scaled to figure a)."

L95f: Why is petrography not important, if mineralogy can have an effect on the surface roughness? Can you at least give the main minerals and which worked well, since this plays a role later? You could also give a reference to some mineral hardness data. It is true that the surface roughness can be affected by the mineralogy. However, the marker structures adhere well to the entire sample, which contains a mixture of minerals typical for a basalt. We therefore do not distinguish between which minerals are suitable for mask structures and which are not. The total assemblage works well, as long as it is properly polished according to geological thin section polishing standards.

Fig. 4: Add the UV-arrows to the legend. F: You might want to put a semicolon after lift-off. This has been adjusted.

L110: For a person not familiar to the process the term "developed" might need an explanation, which – I believe – follows, but it is not clearly connected to the development. An explanation has been added. (now L111)

"Last, the samples are developed by immersing them in a chemical solution that dissolves short polymer chains much faster than long polymer chains, ..."

L116: Does the Ar-plasma not heat the sample? The sample holder is cooled with water to keep the temperature of the substrate low enough. While there is no temperature sensor available in the particular setup used in this work, we know from other setups where a sensor is included on the sample holder that the sample holder's temperature does not heat up to temperatures over 50°C. To give a more specific upper bound for the sputter setup used in this work, we note that another common resist used is PMMA (polymethacrylate). PMMA has a glass temperature of 105°C, which means that if the temperature of the sample would exceed that temperature, the resist would start to flow and features printed into the resist by lithography would be compromised. We do never observe this effect (even for much longer exposure to the Ar-plasma in longer etching/deposition processes). The magnetic interference of the heating is therefore considered to be negligible.

L168: If LED and QDM share their coordinate system and nanoCT are co-registered to QDM, why is co-registration still necessary? Please explain. The LED of the QDM and the magnetic surface map of the QDM share a coordinate system, because they are obtained with the same camera. We co-register the CT data to the QDM data, which can be complicated. Therefore we need the marker structures. This has been clarified in the text.

"Since the LED images of the QDM share their coordinate system with the magnetic surface maps made with the QDM, ..."

L175: Does "thickness" refer to the depth of the feature or to the width on the 2D surface of the sample? This should indeed be 'width' and has been adjusted in the text.

L170: "we are now experimenting" change to "this work provides results on the experiments..." This has been adjusted in the text.

L174f and Fig. 5a-b: Could you shed some light on how you obtained the SEM images? Normally BSE or SE images are in grey scales, so what do the colors mean? To me 5b) looks like a reflected light image. This is indeed incorrect. These are optical microscopy images; this has been adjusted in the text.

L199f: You say that a thicker layer of Nb was not attempted because of the expected problems, but in line 203f you claim that you expect to successfully apply thicker structures. Please explain this contradiction. Why was no attempt made with thicker markers afterwards? There are practical problems in applying a thicker Nb-layer. Since it is quite a complicated and time-consuming procedure to apply the marker structures, we stayed on the safe side for this case study and only tried a thinner layer. However, because we experienced no problems whatsoever during the procedure, and because of the expertise and skills of the lab operator, we expect thicker layers to be possible with a bit of procedure optimization. This lies outside the scope of the current study. This has been clarified in the text.

"A thicker layer was not attempted in this test case, because thicker layers can cause problems during the lift-off phase of the procedure. For example, the structures can collapse or detach from the sample surface. Ideally, we would like to apply a Nb-layer of >400 nm to ensure their visibility in the nanoCT scans, because ca. 400 nm is the CT resolution achieved for standard MMT samples with a

diameter of 1.2-1.5 mm. Although this is a significant increase from the current Nb-layer thickness, we encountered no issues during the lift-off process. Therefore, we expect that thicker marker structures can be successfully applied for future purposes with minor procedure optimization."

L214: "High" resolution normally refers to small pixel/voxel size, but I believe what you mean here is low resolution with high=large pixel size. This is indeed incorrect and has been adjusted in the text.

L214f: How successful was the co-registration? Is the 50-µm marker structure enough? What improved compared to the conventional method? How can you quantify the improvement for such a tedious process? I would have expected a more critical assessment of the results showing e.g. a comparison of co-registration with and without using the markers. Than you could quantify the shift in pixels and show the results to the reader. With the current images this is not possible for the reader. The co-registration can be successfully achieved with the applied marker structures with a 50 µm width. When multiple structures are applied at known distances from one another, they provide a highly reliable reference system for aligning the datasets. The main improvement is that the co-registration process no longer depends on the presence of natural features on the samples, that are recognizable in both types of imaging. This ensures that co-registration is possible for every sample, including those where natural, recognizable features are insufficient. It is difficult to quantify the improvement in terms of pixel shift, because the outcome of co-registration is basically binary: it is possible, or it is not. This has been added to the text. (L221-L228)

"The co-registration can be successfully achieved with the applied marker structures with a 50 µm width. When multiple structures are applied at known distances from one another, they provide a highly reliable reference system for aligning the datasets. The main improvement is that the co-registration process no longer depends on the presence of natural features of the samples that are recognizable in both types of imaging. Instead, the marker structures provide a consistent reference, independent of the samples properties. This ensures that co-registration is possible for every sample, including those where natural, recognizable features are insufficient. Such cases include samples with tiny grain sizes that are difficult to distinguish optically, samples that appear too dark in the LED image of the QDM, or samples with high magnetic mineral density, obscuring the identification of any distinct features."

L55, L57, L173, L240: These suggestions have been implemented where needed.

**RC2**

1) My main concern is related to the results and discussion section. The results section is very short. Can you elaborate more on how efficient this technique was? Furthermore, was registration improved by the marker? The co-registration can be successfully achieved with the applied marker structures with a 50  $\mu$ m width. When multiple structures are applied at known distances from one another, they provide a highly reliable reference system for aligning the datasets. The main improvement is that the co-registration process no longer depends on the presence of natural features on the samples, that are recognizable in both types of imaging. This ensures that co-registration is possible for every sample, including those where natural, recognizable features are insufficient. This has been added to the text. (L221-L228)

"The co-registration can be successfully achieved with the applied marker structures with a 50 µm width. When multiple structures are applied at known distances from one another, they provide a highly reliable reference system for aligning the datasets. The main improvement is that the co-registration process no longer depends on the presence of natural features of the samples that are recognizable in both types of imaging. Instead, the marker structures provide a consistent reference, independent of the samples properties. This ensures that co-registration is possible for every sample, including those where natural, recognizable features are insufficient. Such cases include samples with tiny grain sizes that are difficult to distinguish optically, samples that appear too dark in the LED image of the QDM, or samples with high magnetic mineral density, obscuring the identification of any distinct features."

- 2) Can you elaborate more on the need for this complex method? For which type of samples can it be important? You explain it in the introduction, but it would be good if you could add some examples. This has been added to the discussion. (L225-L228)
- 3) How realistic do you think that this technique will become accessible for people in the field? Is there easy access? Microlithography is a well-established technique in fields such as the semiconductor industry. As demonstrated in our study, laboratories equipped for this type of research can readily apply the method to geological samples as well. In terms of accessibility, the costs are not exceedingly high when compared to the costs of high-end microscopy such as SEM or nanoCT, with which this approach is intended to be combined for co-registration. If no such facilities are available to a user in the field, alternative options are available as outlined in the discussion, although they come with their own limitations.
- 4) Wouldn't a proper calibration of the instruments already solve a lot? The instruments used for the different imaging techniques in this study are fundamentally different. The QDM is camera-based and the CT scanner relies on an X-ray detector. Although both instruments are properly calibrated, they acquire data in very different ways. In practice, this inevitably leads to mismatches in field of view and resolution, complicating accurate co-registration.
- 5) Can you elaborate on the heating step? Is it a useful technique if it is harmful to the magnetic signal? Did you check if this affected your measurements? Generally, heating below 100°C does not harm the magnetic signal, as this temperature is typically below the temperature range of interested in paleomagnetic research. For example, samples in our lab are routinely heated to this temperature to remove any surface contamination. However, if the low temperature magnetic behavior of a sample is the point of interest, it is important to consider the potential effects of heating. This has been clarified in the text. (L93-L95)

"Another important consideration is that the samples are subjected to a <100°C heating step during the procedure, which may interfere with the magnetization of the sample. However, this is generally not problematic in paleomagnetic research, since the low-temperature magnetic signal is rarely the point of interest."

- 6) Do you know the source power? That is more commonly used than Source Current. The table lists the parameters that can be adjusted as part of the measurement configuration. These include source current and source voltage.
- 7) You need to be careful with the resolution and voxel size, as they are not the same. If the resolution is 1.38  $\mu$ m, you will be able to see an object 3  $\mu$ m wide. It is indeed true that resolution and voxel size are not the same, this has been clarified in the text.

Minor textual changes have been made and are indicated directly in the enclosed pdf for your convenience. We want to thank the reviewers again for helping us in improving this manuscript.

On behalf of all authors,

Rosa de Boer

Postdoctoral researcher at paleomagnetic laboratory Fort Hoofddijk, Utrecht University, NL